complexity

agglomeration economies, urban scaling, urbanization and localization economies, occupational concentrations

**Author for correspondence:**
Somwrita Sarkar
e-mail: somwrita.sarkar@sydney.edu.au

# Evidence for localization and urbanization economies in urban scaling

Somwrita Sarkar[1], Elsa Arcaute[2], Erez Hatna[3], Tooran Alizadeh[1], Glen Searle[1] and Michael Batty[2]

[1]School of Architecture, Design, and Planning, The University of Sydney, Sydney 2006, Australia
[2]Centre for Advanced Spatial Analysis, University College London, 90 Tottenham Court Road, London W1T 4TJ, UK
[3]School of Global Public Health, New York University, 715/719 Broadway, New York, NY 10003, USA

SS, 0000-0002-2037-0979; MB, 0000-0002-9931-1305

We study the scaling of (i) numbers of workers and aggregate incomes by occupational categories against city size, and (ii) total incomes against numbers of workers in different occupations, across the functional metropolitan areas of Australia and the USA. The number of workers and aggregate incomes in specific high-income knowledge economy-related occupations and industries show increasing returns to scale by city size, showing that localization economies within particular industries account for superlinear effects. However, when total urban area incomes and/or gross domestic products are regressed using a generalized Cobb–Douglas function against the number of workers in different occupations as labour inputs, constant returns to scale in productivity against city size are observed. This implies that the urbanization economies at the whole city level show linear scaling or constant returns to scale. Furthermore, industrial and occupational organizations, not population size, largely explain the observed productivity variable. The results show that some very specific industries and occupations contribute to the observed overall superlinearity. The findings suggest that it is not just size but also that it is the diversity of specific intra-city organization of economic and social activity and physical infrastructure that should be used to understand urban scaling behaviours.

# 1. Introduction

Urban scaling relationships summarize how the size of a city, usually measured by its total population size, can be used as a

predictor for its socio-economic and spatial characteristics [1,2]. The standard urban scaling model used in this domain takes the following form:

$$y = \alpha X^\beta,$$ (1.1)

where $y$ is an urban indicator, $X$ is the size of the city, usually represented as the residential population of a city, $\alpha$ is a constant of proportionality and $\beta$ is the scaling exponent. One of the most common uses of $y$ is as a productivity indicator, where $y$ represents income, gross domestic product (GDP) or wages. Thus, this scaling model can be used to evaluate how the productivity of a city varies with the size of its population, and how big a role does city size play in explaining overall productivity of a city. It has been shown in previous research that the above relationship between urban size and productivity in cities across several countries including the USA, China and Germany manifests a superlinear increase of total economic output against city size, or increasing returns to scale [1]. However, other studies have shown that variations on the definitions of what a 'city' is can show a linear increase of total economic output against city size, or constant returns to scale [3].

The general form of the scaling function derives from a more general two-factor Cobb–Douglas production function [4]

$$y = \alpha L^\beta K^{1-\beta},$$ (1.2)

where $L$ represents labour input, $K$ represents capital input, $\alpha$ could be a constant of proportionality or it could take a more complex form as a function of the population, i.e. $\alpha = g(X)$. If the constraint that the exponents should add up to 1 is removed, then the form would be $y = \alpha L^{\beta_1} K^{\beta_2}$, where $\beta_1 + \beta_2$ could add up to any value (less than 1, equal to 1 or more than 1). Output $y$ is then dependent on the factors of production $L$ and $K$, and on $\alpha$ as a function of population.

For the Cobb–Douglas form, similar to the urban scaling form, if the exponents sum to 1, a linear or constant returns to scale is assumed; if the exponents sum to a value between 0 and 1, a sublinear or decreasing returns to scale relationship is assumed; if they sum to greater than 1, a superlinear or increasing returns to scale relationship is assumed. Parallels between urban scaling behaviour and Cobb–Douglas formulations have been explored in previous papers [2,4,5].

Another series of papers from the 1960s to the 1980s also explore the relationship between the productivity of a city as a function of its size, using generalized Cobb–Douglas functions [6–10]: output is dependent on a number of inputs, most commonly capital, labour and city size, but organized into finer distinctions of capital and labour. The production functions studied are generalized Cobb–Douglas of the form

$$y = g(X)f(L, K),$$ (1.3)

where the term $g(X)$ captures the effects of population size, and the term $f(L, K)$ captures the effects of labour and capital inputs. These individual terms are complex in themselves and capture multiple types of input from population size, labour and capital (for example, regional population effects, multiple types of capital inputs coming in from plants or physical infrastructure, or multiple types of labour inputs, coming in say from different industry types). The terms $g(X)$ and $f(L, K)$ can be modelled mathematically using the scaling idea that each input component explaining overall productivity scales with a different exponent

$$y = \alpha X^{\beta_0} \prod_j L_j^{\beta_j} \prod_m K_m^{\beta_m},$$ (1.4)

where $X^{\beta_0}$ as before captures the effects of population, and a number of $L$ and $K$ inputs could be used, each with its own exponent. In a log-transformed linear regression of the above, the estimated coefficients $\beta_0$, $\beta_j$ and $\beta_m$ would then reveal the importance of population size, or specific labour and capital inputs, respectively, in explaining total productivity or output $y$.

While it would be ideal to have a simpler model where fewer inputs could lead to the best fit, the possibility that labour and capital variations in combinations, covering factors such as industrial specialization and diversity, could explain productivity in addition to population size should also be considered. One could then check whether population size remains significant and relevant as an explanatory variable as the models grow in complexity and numbers of inputs. All these historical studies collectively find that population size alone does not explain productivity differentials, and that labour and capital variations significantly account for regional variations of productivity [6–8]. Moomaw [7,8] finds that doubling population results in a 1.5% rise in productivity, while Hyclak [6] finds this value to be 4.5%, with a range of other values reported from other studies. The effect of

population size in these estimates is much lower than the findings in [1,2,4], where only a single factor input, namely population size, is considered. Furthermore, in the more detailed models, it is found that capital and labour have statistically significant and stronger effects than population size alone, signalling that types of occupations and industries as well as types of capital inputs can affect the productivity significantly. Overall, these findings show that local conditions (that appear as heteroskedastic fluctuations around the average in the scaling law in equation (1.1)) could significantly vary depending on specific industry or population mixes, and can affect overall productivity for a city. In particular, some authors have shown that the maturity of the specific industry is a more relevant determinant for the exponent than population size, and connect the value of scaling exponents with the stage of activities in a hierarchical diffusion process of innovation waves within the system of cities within a geographical evolutionary theory of urban systems framework [11–15].

Rosenthal & Strange [16] briefly note that while agglomeration economies are particularly pronounced as localization economies (economies of scale arising within particular industries), they are much less pronounced and not strong at the level of urbanization economies at the metropolitan level (economies of scale due to city size). This implies that empirical evidence on the scaling behaviour of particular industries and occupations should be studied along with city-level or metropolitan-level scaling behaviour, something that also emerged from previous findings [17].

The observed superlinear growth of economic output against city size has also been explored from several other parallel directions. One such direction established spatial aggregation as an important factor, where it was shown that when urban area boundaries or city definitions are varied, many urban indicators that show superlinear scaling under one definition end up showing linear scaling under other varying spatial aggregation schema [3]. Similarly, varying urban area definitions showed that scaling of $CO_2$ emissions by city size could fall in the superlinear or sublinear regime, dependent on adopting two different types of metropolitan area definitions [18]. Furthermore, assuming different generative models for the data, it was shown that alternative functional forms could well explain the observed scaling behaviours [17,19].

Another direction of inquiry brings into focus the question of distribution as an important factor. While scaling laws in their current form consider only aggregative characteristics, in the form of total incomes or wages, total GDPs, or other such aggregated values measured against total population size, it was shown that when distributions are considered, different parts of the income distribution or housing cost distributions may scale in different regimes [20,21]. In particular, while lower income and housing cost categories were shown to scale sublinearly or linearly with city size, highest housing costs and income categories emerged as scaling superlinearly, thereby showing an emergent distributional inequality in which larger cities disproportionately agglomerate the richest sections of the population. This result shows that while aggregate quantities show an overall scaling, parts of the distributions of the aggregate may show differential scaling in all three regimes, sublinear, linear or superlinear. A similar observation was also reported with different industry types at different stages of evolution in a separate geographical context [13].

In the present paper, we focus on a third direction (different from spatial aggregation or income/living cost category aggregation): this is on sectoral and occupational aggregation. This direction delves further into the distributional questions [13,20,21] of how differential scaling of productivity may emerge depending on occupational categories or industry sectors [16,22,23].

The rest of the paper is organized as follows. Section 2 provides evidence for the existence of localization economies, as when aggregate income outputs or number of workers from different occupations are regressed against total population size of an urban area, evidence for differential scaling may be seen: lower- or middle-income occupations scale very differently from high-income occupations. For certain very high-income occupations, there is a critical size below which they do not occur, showing that specific economic functions emerge in cities beyond an agglomeration threshold, supporting recent observations on the presence of valuable information measured through the absence of information [24].

Section 3 demonstrates the parallels of mathematical form between urban scaling functions and Cobb–Douglas production functions. It then uses a generalized urban production function in which total economic output (total income, wages, GDP etc.) is modelled as a function of total population size of a city as well as several other input factors, such as workers employed in different occupational categories or industrial categories.

In §3 also, urbanization economy effects are shown. The coefficients of this more generalized production function are estimated using data from Australia and the USA, and it is shown that when populations are disaggregated in the form of labour input in this way, the total output shows constant returns to scale.

Section 4 offers an overall critique of urban scaling laws and the production function connection, reflects on the current state of research, and establishes some open questions in this line of research. In particular, it is established that while these systematic mathematical relationships exist and show empirical robustness, generative models of underlying processes explaining these empirical regularities are missing.

Two principal observations emerge. First, size is not the only determinant of economic and social performance. In particular, productivity is a function also of industry and occupational organization or functions. The variance observed in urban scaling laws (cities of the same size showing fluctuations around the mean expected behaviour) can be explained by the specifics of these, since different industries and occupations are likely to be organized through different agglomeration economies (localization and urbanization), and specific combinations of these in each individual city would then govern city-wide behaviour.

Second, the results show that localization economies may show increasing returns to scale, but urbanization economies may show constant returns to scale. Thus, the observation that specific cities or city systems follow increasing returns to scale may well be a consequence of some of the industries, which are themselves organized around increasing returns, being disproportionately present in these cities. For the same reason, some small cities that agglomerate particular industries where increasing returns operate emerge as outliers or large heteroskedastic fluctuations in the overall scaling plot [21].

# 2. Examining localization economies: scaling of number of workers and total income in specific occupations

In this section, we study scaling of the aggregate numbers of workers and the aggregate incomes in different occupational categories against total population size for Australian urban areas, which are called significant urban areas (SUAs). The basis for using SUAs as a functional metropolitan region has been discussed in [20]. The SUAs range from urban denoted populations of 10 000 up to 5 million. We find that larger cities, in general, attract more workers in the highest paid occupations and industries.

## 2.1. Total personal income against city size

First, we study the scaling of total personal income against city population size. The total personal income earned in 2015–2016 is extracted per statistical area level 2 (SA2) and then aggregated to the SUA level (SA2s are the smallest statistical geography-based areas for which complete Census data is available in Australia). This total income is a measure of income from all persons who submitted a tax return, and includes income from employee income, income from own unincorporated business, income from investment and income from pensions and annuities.

Using the scaling form in equation (1.1), we first ran an overall regression for total personal income (employee, own unincorporated business, investment, and superannuation and annuities) against total population size. All linear regressions in the manuscript were performed using the Matlab regression package *fitlm*. For the maximum-likelihood estimation (MLE) methods, the original code provided by the authors of [19] was used.

For the overall regression on total income, the value of the scaling exponent $\beta$ was 1.05 (Adj $R^2 = 0.98$) by the ordinary least-squares (OLS) method. Through the MLE method [19], the estimate was for a scaling model estimate of $1.05 \pm 0.01$. This is mildly superlinear, and less than the estimates for the USA, which were reported as close to 1.12–1.15 [1,4]. When total income was divided by the total number of workers to compute income per worker, however, the $\beta$ estimate was only 0.02 (Adj $R^2 = 0.02$). This observation, that per capita estimates are much noisier than expected when compared to estimates based on aggregate data, was also earlier noted by Shalizi [17].

## 2.2. Total number of workers and total income in occupations by city size

Occupational categories in Australia are defined at four levels with increasing granularity of job definitions: OCCP1 (eight categories), OCCP2 (51 categories), OCCP3 (134 categories) and OCCP4 (474 categories) following the Australian and New Zealand Standard Classification of Occupations (ANZSCO). These data are extracted using the Census Table Builder facility offered by the Australian Bureau of Statistics.

The main analysis presented in this paper focuses on the OCCP1 and OCCP2 levels. In principle, it is possible to analyse the OCCP3 and OCCP4 levels using exactly the same framework as presented here,

but with further levels of granularity the number of zero or absent values increases (numbers of workers in a certain occupation are missing in a city), and it becomes difficult to decide whether the zero number of workers corresponds to a city-specific characteristic (the city does not specialize in that occupation, but other cities of the same size may have workers in this occupation) or whether the zero value emerges from a size issue (the city's size is predictive of no workers in that occupation since the occupation only occurs beyond a critical size threshold) [24]. An example of the former would be a specific economic activity like mining: some small towns specialize in the activity, but others do not. An example of the latter would be the occupations within the finance industry: the industry itself occurs in large cities because of the ties and dependencies it shares with other industries (e.g. finance, investment banking), such that it would be rare to find its presence in a small city. Thus, the analysis is restricted to OCCP1 and OCCP2 levels, but the observed patterns are sufficiently strong to provide evidence for the claims in the paper.

The number of workers in each OCCP1 and OCCP2 level is tabulated for 101 SUA, which are designated as urban in Australia [20].

To extract total income in each occupation, income data has to be extracted filtered by occupational class. In the 2016 Census, 13 income categories or bands are defined ranging from $1–$149 weekly (or $1–$7799 yearly) to more than $3000 weekly (or $156 000 yearly). Thus, data can be extracted per SUA for the count of the number of workers in each income category.

Using the data described above, the aggregate number of workers per occupational category per SUA is counted. Furthermore, total aggregate income in a single occupation is measured by counting the number of people employed in that occupation per income category, multiplying by the income earned in each of these categories and then adding them

$$I_c = \sum_{i=1}^{m} n_i a_i,$$ (2.1)

where there are $m$ income categories, $n_i$ is the number of people in an occupation in income category $i$ and $a_i$ is the median or mean income earned in a single income category. Then, $I_c$ is the total income defined in a single occupation and this value can be computed for all the different $n$ occupation categories, $c = 1 \ldots n$.

Then, for the sectorally disaggregated analysis, in a first set of regressions we let $y$ represent the total number of workers in an occupation, and study the scaling of this $y$ against city size for all the $n$ occupation categories. In a second set of regressions, we let $y$ represent the total aggregate income earned in an occupation, and study the scaling of this $y$ against city size for all the $n$ occupation categories. In all these sets of regression, we estimate the value of the exponent $\beta$.

We compute the $\beta$ estimates using both (i) OLS regression of the log-transformed model of the scaling equation $\ln y = \ln \alpha + \beta \ln X$, and (ii) a generalized MLE-based optimization (with the constant variance assumption or the homoskedasticity assumption in OLS relaxed to allow for other likelihood functions) in which two models are compared: a lognormal model (representing the scaling model) and a fixed $\beta = 1$ linear model. If either of the models are statistically significant, then we choose $\beta$ from that model. If both models are not significant, then we choose the $\beta$ from the model with the least Bayesian information criteria (BIC) statistic [19]. The application details of the MLE-based process are described in detail in [21] using the framework proposed in [19]. In general, the estimates from OLS and MLE were similar, but OLS estimates ranging from 0.98 to 1.02 were estimated as linear (i.e. the linear model was preferred over the scaling model) by the MLE approach.

Tables 1 and 2 show the scaling exponents for the number of workers and the aggregate income per occupation in occupational levels 1 and 2 (OCCP1 and OCCP2) categories, respectively. In the tables below, an income cluster label is computed, to identify low/medium/high-income occupations. The occupations are classified into low-, medium- and high-income occupations in order to explore whether there is a correlation between a specific occupation, its broad income categorization, and its scaling characteristics. This is an important point to explore because we have found in previous work [20] that higher incomes scale superlinearly by city size, while lower and medium incomes scale linearly or sublinearly. Since incomes earned are correlated to the specific occupations in which they are earned, it could well be the case that the scaling of incomes ultimately ties to the scaling of specific occupational categories. For example, it is well known that knowledge- or finance-based occupations are typically high income and are seen mostly in the largest cities.

The labels are computed using a spectral clustering algorithm for bipartite networks/data, using the singular value decomposition [25], where the OCCP1 or OCCP2 categories form the rows, and the

**Table 1.** Scaling exponent values estimated for aggregate numbers of workers and total income in OCCP1 categories against population size. Results from both the OLS and the MLE approach are presented. The $\beta$-estimate column specifies whether the lognormal model is chosen, or the linear model is chosen in each case. The income cluster label identifies whether worker distributions in each occupation by income bands lead to an occupation being classified as a high or medium-low income occupation through clustering analysis. The colour in the final column shows the 'high' and 'medium-low' category clusters for the different occupation classes.

| occupations (OCCP1) total number of workers in each category | $\beta$ estimate (OLS estimate) [Adj $R^2$] (aggregate workers) | $\beta$ estimate ± error (MLE estimate)* (aggregate workers) | $\beta$ estimate (OLS) [Adj $R^2$] (aggregate income) | $\beta$ estimate ± error (MLE estimate)* (aggregate income) | income cluster |
|---|---|---|---|---|---|
| managers | 1.07 [0.98] | 1.07 ± 0.01 (lognormal) | 1.11 [0.97] | 1.11 ± 0.02 | high |
| professionals | 1.13 [0.98] | 1.12 ± 0.03 (lognormal) | 1.15 [0.97] | 1.14 ± 0.02 | high |
| technicians and trades workers | 0.99 [0.98] | 1.00 (fixed $\beta = 1$) | 0.99 [0.93] | 1.00 (fixed beta) | medium-low |
| community and personal services workers | 1.02 [0.99] | 1.00 (fixed $\beta = 1$) | 1.01 [0.98] | 1.00 (fixed beta) | medium-low |
| clerical and administrative workers | 1.08 [0.99] | 1.07 ± 0.03 (lognormal) | 1.10 [0.98] | 1.10 ± 0.05 | medium-low |
| sales workers | 1.02 [0.99] | 1.00 (fixed $\beta = 1$) | 1.05 [0.99] | 1.05 ± 0.01 | medium-low |
| machinery operators and drivers | 0.93 [0.91] | 1.00 (fixed $\beta = 1$) | 0.90 [0.82] | 0.90 ± 0.02 | medium-low |
| labourers | 0.95 [0.98] | 0.95 ± 0.02 (lognormal) | 0.95 [0.96] | 0.95 ± 0.02 | medium-low |

income bands form the columns, and each matrix entry is a count of the number of workers in each occupation in each income level for the whole of Australia. The clustering shows two principal clusters for the OCCP1 level, corresponding to high and medium-low incomes, and three principal clusters for the OCCP2 level, corresponding to high, medium and low incomes. In some cases, category splits occur. For example, while machinery operators and drivers at the OCCP2 level are a medium-low category occupation, at the OCCP3 level a split of this category, machinery and stationary plant operators move to the high-income cluster, while mobile plant operators move to the medium income cluster. This could be due to the higher skills required in plant operators within specific industries (e.g. mining and metallurgical processes), where higher specialized sets of skills could be correlated with higher returns.

Table 1 shows conclusively that the number of workers as well as the incomes in the highest paid knowledge economy occupations (managers and professionals) are superlinear with city size, whereas the lower paid occupations are either linear or sublinear. The only category of low-paid occupations that is superlinear in both the number of workers as well as the income earned is clerical and administrative workers, who would be in supporting roles to managers and professionals. The category of community and personal services workers is linear in the number of workers, but is slightly superlinear in the income earned.

Table 2 shows the same pattern as in table 1, but with finer granularity. In particular, it is very clear scaling parameters for the large variety of medium- and low-paid service provision occupations are sublinear or linear in both the number of workers as well as the aggregate incomes in that category.

**Table 2.** Scaling exponent values estimated for aggregate numbers of workers and aggregate income in OCCP2 categories against population size. Results from only the OLS regression are presented, since these are very close to the MLE results. The colour in the final column shows the 'high', 'medium' and 'low' category clusters for the different occupation classes. The orange colour in the other columns shows all the exponent beta in the scaling analysis that are above a value of 1.0.

| OCCP2 categories | $\beta$-estimate (OLS) aggregate workers | Adj $R^2$ | $\beta$-estimate (OLS) aggregate income | Adj $R^2$ | income cluster |
|---|---|---|---|---|---|
| chief executives, general managers and legislators | 1.17 | 0.95 | 1.19 | 0.94 | high |
| specialist managers | 1.14 | 0.97 | 1.16 | 0.96 | high |
| business, human resource and marketing professionals | 1.23 | 0.97 | 1.25 | 0.96 | high |
| design, engineering, science and transport professionals | 1.16 | 0.93 | 1.15 | 0.90 | high |
| education professionals | 1.04 | 0.98 | 1.04 | 0.97 | high |
| health professionals | 1.09 | 0.97 | 1.11 | 0.96 | high |
| ICT professionals | 1.47 | 0.93 | 1.43 | 0.93 | high |
| legal, social and welfare professionals | 1.11 | 0.96 | 1.14 | 0.95 | high |
| engineering, ICT and science technicians | 1.07 | 0.94 | 1.05 | 0.88 | high |
| electrotechnology and telecommunications trades workers | 1.00 | 0.94 | 0.99 | 0.88 | high |
| protective service workers | 1.04 | 0.89 | 1.02 | 0.88 | high |
| office managers and programme administrators | 1.10 | 0.98 | 1.14 | 0.96 | high |
| sales representatives and agents | 1.14 | 0.97 | 1.16 | 0.96 | high |
| machine and stationary plant operators | 0.85 | 0.70 | 0.81 | 0.59 | high |
| farmers and farm managers | 0.70 | 0.46 | 0.65 | 0.57 | medium |
| hospitality, retail and service managers | 1.02 | 0.99 | 1.04 | 0.98 | medium |
| arts and media professionals | 1.21 | 0.96 | 1.16 | 0.95 | medium |
| automotive and engineering trades workers | 0.89 | 0.88 | 0.89 | 0.79 | medium |
| construction trades workers | 1.07 | 0.95 | 1.08 | 0.94 | medium |
| food trades workers | 1.02 | 0.98 | 1.01 | 0.98 | medium |
| skilled animal and horticultural workers | 0.99 | 0.95 | 0.96 | 0.95 | medium |
| other technicians and trades workers | 1.03 | 0.98 | 1.02 | 0.93 | medium |
| health and welfare support workers | 0.97 | 0.96 | 0.96 | 0.96 | medium |
| personal assistants and secretaries | 1.10 | 0.97 | 1.10 | 0.97 | medium |
| general clerical workers | 1.05 | 0.98 | 1.06 | 0.96 | medium |
| inquiry clerks and receptionists | 1.08 | 0.98 | 1.08 | 0.98 | medium |
| numerical clerks | 1.09 | 0.98 | 1.12 | 0.98 | medium |
| clerical and office support workers | 1.04 | 0.98 | 1.02 | 0.97 | medium |
| other clerical and administrative workers | 1.10 | 0.96 | 1.09 | 0.95 | medium |
| mobile plant operators | 0.93 | 0.91 | 0.90 | 0.89 | medium |
| road and rail drivers | 0.96 | 0.96 | 0.94 | 0.91 | medium |
| storepersons | 1.17 | 0.92 | 1.07 | 0.90 | medium |
| construction and mining labourers | 1.00 | 0.95 | 1.00 | 0.91 | medium |
| factory process workers | 1.01 | 0.72 | 0.97 | 0.72 | medium |

**Table 2.** (Continued.)

| OCCP2 categories | β-estimate (OLS) aggregate workers | Adj $R^2$ | β-estimate (OLS) aggregate income | Adj $R^2$ | income cluster |
|---|---|---|---|---|---|
| carers and aides | 0.99 | 0.99 | 0.98 | 0.98 | low |
| hospitality workers | 1.07 | 0.97 | 1.04 | 0.97 | low |
| sports and personal service workers | 1.11 | 0.98 | 1.10 | 0.97 | low |
| sales assistants and salespersons | 0.99 | 0.99 | 1.00 | 0.99 | low |
| sales support workers | 1.04 | 0.99 | 1.01 | 0.98 | low |
| cleaners and laundry workers | 0.95 | 0.98 | 0.96 | 0.97 | low |
| farm, forestry and garden workers | 0.80 | 0.81 | 0.77 | 0.84 | low |
| food preparation assistants | 1.00 | 0.98 | 0.95 | 0.98 | low |
| other labourers | 0.96 | 0.97 | 0.94 | 0.93 | low |

Overall, the results show that within the knowledge industry categories and supporting occupational categories, there is a superlinear effect observed against city size: numbers of workers and income earned shows increasing returns to scale by city size. The medium- and low-income occupation clusters do not, on the whole, show this effect. To the contrary, they only scale linearly or sublinearly with city size. This provides evidence for localization economies operating in high-income occupations: when these specific occupations agglomerate in a particular city, a large part of the total output or income of the city will come from these occupations. Furthermore, the results show that these industries show a tendency to agglomerate in the largest cities, and thus, the larger the city, the higher the proportion of its overall income that is earned in these occupations.

# 3. The relationships between a generalized production function and scaling laws

The standard form for urban scaling functions takes the following form as stated in equation (1.1), repeated here for convenience:

$$y = \alpha X^{\beta}, \tag{3.1}$$

where as before $y$ is total income earned in an urban area, and $X$ is city population. We note here the similarity of this form with a Cobb–Douglas production function (equation (1.2)) which we also restate here

$$y = \alpha L^{\beta} K^{1-\beta}, \tag{3.2}$$

where $L$ represents labour input and $K$ represents capital input, and the exponents sum to 1 (assuming constant returns to scale). Taking logs on both sides gives us

$$\ln y = \ln \alpha + \beta \ln L + (1 - \beta) \ln K, \tag{3.3}$$

and $\beta$ can be estimated from a linear least-squares regression. Following previous practice [6–10], we consider a generalized form of the production function as

$$y = \alpha \prod_{i=1}^{n} X_i^{\beta_i}, \tag{3.4}$$

where $X_i$ represents the number of people in occupational sector $i$, and $\beta_i$ represents the exponents for the different growth rates (or elasticities) for the different occupational categories. In principle, total population $X$ is a sum of the total number of workers in each occupational category ($X_i$) and everyone else who is not in the labour force ($X_0$), so that $X = \sum_{i=1}^{n} X_i + X_0 = \sum_{i=0}^{n} X_i$. Furthermore, if all the exponents $\beta_i$ sum to 1, then $y$ will show constant returns to scale; if all the exponents $\beta_i$ sum to less than 1, then $y$ will show decreasing returns to scale; and if all the exponents $\beta_i$ sum to greater than 1, then $y$ will show increasing returns to scale.

**Table 3.** Scaling exponent values estimated for aggregate numbers of workers in OCCP1 categories against total income in the urban area.

| $X_i$ | $\beta$ estimate $R^2 = 0.99$ | $p$-value | $X_i$ | $\beta$ estimate $R^2 = 0.99$ | $p$-value |
|---|---|---|---|---|---|
| $X_0$ | 0.00 | 0.99 | SUA total population | 0.07 | 0.49 |
| $X_1$: managers | 0.23 | 0.00 | $X_1$: managers | 0.24 | 0.00 |
| $X_2$: professionals | 0.24 | 0.00 | $X_2$: professionals | 0.24 | 0.00 |
| $X_3$: technicians and trades workers | 0.59 | 0.00 | $X_3$: technicians and trades workers | 0.58 | 0.00 |
| $X_4$: community and personal services workers | 0.00 | 0.99 | $X_4$: community and personal services workers | −0.02 | 0.77 |
| $X_5$: clerical and administrative workers | 0.12 | 0.13 | $X_5$: clerical and administrative workers | 0.12 | 0.16 |
| $X_6$: sales workers | −0.29 | 0.00 | $X_6$: sales workers | −0.32 | 0.00 |
| $X_7$: machinery operators and drivers | 0.10 | 0.00 | $X_7$: machinery operators and drivers | 0.10 | 0.00 |
| $X_8$: labourers | 0.01 | 0.82 | $X_8$: labourers | 0.004 | 0.94 |

Taking logs on both sides of the equation above, we will have

$$\ln y = \ln \alpha + \sum_{i=0}^{n} \beta_i \ln X_i, \tag{3.5}$$

and the values of the exponents $\beta_i$ can now be estimated using multivariate linear regression on this log-transformed model.

We run two regressions, first with the number of workers in each OCCP1 category

$$\ln y = \ln \alpha + \beta_0 \ln X_0 + \sum_{i=1}^{n} \beta_i \ln X_i, \tag{3.6}$$

with $X_0$ representing the number of people who are not in the labour force, and $X_i$, $i = 1 \dots n$ representing the eight OCCP1 categories. The first two columns of table 3 show the results.

A second regression is run with the number of workers in each OCCP2 category, with $X_i$, $i = 1 \dots n$ representing the 43 OCCP2 categories. Here, $X_0$ represents the number of people who are not in the labour force, and $X_i$, $i = 1 \dots 43$ represents the 43 OCCP2 categories.

For both cases, some clear results and conclusions emerge

  (i) The exponent $\beta_0$ for the OCCP1 case comes out to be virtually zero and not significant (as it has a large $p$-value), and for the OCCP2 case comes out to be 0.02 and again not significant.
 (ii) The sum of the exponents for the OCCP1 case comes out to be $\sum_{i=1}^{8} \beta_i = 0.998 \approx 1.00$ (results shown in the first two columns of table 3) and the sum of the exponents for OCCP2 case comes out to be $\sum_{i=1}^{43} \beta_i = 0.977 \approx 1.00$.
(iii) If we replace $X_0$ as the total SUA population, the sum of the exponents is still approximately 1.00, with the coefficient $\beta_0 = 0.07$, but is not significant (large $p$-value) in the OCCP1 case, (results shown in the last two columns of table 3), and $\beta_0 = 0.12$ in the OCCP2 case and not significant. Furthermore, the coefficients for multiple occupational categories come out to be higher than the population size coefficients and are significant.

These results thus show that sorting is more important than overall population size in explaining productivity scaling. The scaling comes from population sorting (i.e. larger cities have more of certain high-income industries but the industries are the same everywhere they exist). The $\beta$s that are larger than 1 for particular industries are balanced by those in industries below 1, thereby bringing the

overall scaling into the linear range. Second, in each regression above, the sum of the exponents together come out to be in the linear range and not in the superlinear range.

In [17], the author showed an effect similar to point (i) above by considering a log-additive model and considering industry shares as inputs, where it was shown that the explanatory power of the total population term became much smaller when industry sectors were considered. Here, in addition to such an effect, we also find that the overall function scales with constant returns to scale, and not increasing returns to scale.

We re-perform this analysis for data from the USA considering the metropolitan statistical areas (MSA), with GDP per MSA as the output variable $y$, and inputs as total population $X_0$, and numbers of workers in all the 13 industrial categories as follows:

—$X_0$: total population
—$X_1$: agriculture, forestry, fishing and hunting, and mining
—$X_2$: construction
—$X_3$: manufacturing
—$X_4$: wholesale trade
—$X_5$: retail trade
—$X_6$: transportation and warehousing, and utilities
—$X_7$: information
—$X_8$: finance and insurance, and real estate and rental and leasing
—$X_9$: professional, scientific and management, and administrative and waste management services
—$X_{10}$: educational services, and healthcare and social assistance
—$X_{11}$: arts, entertainment and recreation, and accommodation and food services
—$X_{12}$: other services, except public administration
—$X_{13}$: public administration.

For the USA analysis too, two results emerge:

(i) The exponent $\beta_0$ comes out to be virtually zero, and
(ii) The sum of the exponents comes out to be $\sum_{i=1}^{13} \beta_i = 1.03$.

Thus, while the USA data does show a total exponent value that is slightly higher than 1, this value is much smaller than the $\beta = 1.12$ value that emerges when total GDP is regressed against total population as reported in previous work [1,17]. Furthermore, the explanatory power of the total population in the above regression comes out to be weaker than sorting and industrial composition, which can more strongly explain the GDP output variable.

# 4. Conclusion: overall critiques of urban scaling theory

In this paper, we studied the scaling behaviour of number of workers and aggregate incomes in different occupations and industrial categories against city size, and the scaling behaviour of total incomes against number of workers in different industries. We reflect on the key results here.

First, the number of workers and aggregate incomes in highly paid occupations (usually knowledge economy and supporting occupations) are correlated with city size with superlinear effects being evident. Since knowledge economy occupations tend to cluster in the largest cities, regressions against city size are also likely to show superlinear returns by city size, when in fact the superlinearity may be arising from occupational or industrial organization and not from population size *per se*. For example, a smaller city with very high concentrations of high-income occupations may then emerge as an outlier (as discussed for Cambridge, for example, in [3], or for Australian mining towns in [21]).

Second, when the number of workers are used as labour factor inputs to total income or GDP as output, no strong evidence for superlinear scaling of output is found, instead the evidence seems to tilt towards linear scaling and constant returns to scale. Thus, localization economies for particular industries and occupations, especially knowledge economy ones, show increasing returns, but urbanization economies would tend to show constant returns to scale.

Third, putting together the above two results, one is tempted to conclude that while individual 'organs' of a city may show increasing returns, the city as a whole operates in the constant returns regime. This derives from the observation that while incomes in specific occupations are showing superlinear returns, both in the number of workers as well as the income earned in these specific occupations and industries, when total city income is regressed against labour input, constant returns are observed.

Fourth, the conjecture above would be a premature conclusion, since a further generalized Cobb–Douglas-style function could model capital input components along with labour input components, and it is as yet unknown what the regression results would be in that case. Specifically, it is extraordinarily difficult to build up a capital component and should be the topic of future research, since this would include outputs from the total physical infrastructure of a city, including housing, roads, factories and plants, equipment, new technology/capital investments into cities, and so on.

Finally, in its current form, urban scaling theory is at best a data fitting method, providing only preliminary insight into the causal processes by which such effects are observed. Thus, future advances should look behind causal economic and physical processes that result in observed scaling behaviours.

Data accessibility. The data and code files related to this project are available at: https://doi.org/10.5281/zenodo.3596604 [26].

Authors' contributions. S.S. designed the experiments, performed the analysis and wrote the paper. E.A., E.H. and M.B. assisted with experiment design, analysis and wrote the paper. T.A. and G.S. provided urban planning and policy relevance insights.

Competing interests. We declare we have no competing interests.

Funding. The research and manuscript preparation did not draw on any funding source.

Acknowledgements. S.S. thanks Prof. Mike Batty for hosting her as Academic Visitor at UCL CASA. The work in the manuscript is done as a collaboration between the University of Sydney and UCL CASA as a result of this visit. S.S. thanks David Levinson for helpful and stimulating discussions on the subject matter of the manuscript. The authors thank the two anonymous referees for helpful comments and suggestions.

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
