## [Reviewer comments · Royal Society Open Science]

Review History

RSOS-191638.R0 (Original submission)

Review form: Reviewer 1

Is the manuscript scientifically sound in its present form?

No

Are the interpretations and conclusions justified by the results?

Yes

Is the language acceptable?

Yes

Do you have any ethical concerns with this paper?

No

Have you any concerns about statistical analyses in this paper?

Yes

Recommendation?

Accept with minor revision (please list in comments)

Comments to the Author(s)

Evaluation of “Evidence for localization and urbanization economies in urban scaling”

The authors investigate urban scaling in terms of occupation and corresponding income. Therefore various analysis have been performed mostly for Australia but also for the USA. First, total personal income vs. population leads to slightly super-linear scaling with an exponent 1.05 for Australia. Second, the number of works separated into various occupations is regressed versus city size, resulting in super-linear scaling for high-income occupations and linear or even sub-linear for other occupations (again for Australia). Last, a multi-linear regression in the spirit of Cobb-Douglas Production function is performed providing an estimated exponent for each occupation group. For Australia, the authors find that the exponent of non-working population is 0 and if replaced by the city population it is also close to zero. Moreover, in both cases the sum of the exponents is close 1. Consistent results are obtained for MSA in the USA, although not shown. The authors draw conclusions as summarized in the Abstract (essentially it questions increasing returns to scale of urban economies).

The paper is not only a nice read, but it is also of high relevance for the urban scaling community and for the empirical understanding of agglomeration economies. Therefore, I recommend publication but I also think it would be good to address some issues.

- I have concerns about the multivariate linear regression. First it is not detailed how it has been done methodologically. Second, probably all X_i are correlated which can cause trouble. Third, it is striking that the sum of the exponents is always so close to 1. Can there be a methodological reason? E.g. that the sum converges to 1 when many independent (correlated) variables are used?
- The authors mention the problem of zero numbers in the counts of workers, but how do they deal with it? Or are there no zero-entries, not even in OCCP2.
- The authors employ some clustering. Neither the purpose is explained nor the clustering itself becomes clear.
- As the income is a product of the number of workers, it is expected that the analysis of number of workers and income lead to similar results.
- The authors cite an unusual small number of papers. Although there is no lower limit, I still suggest to expand the citations. E.g. recently H.V. Ribeiro et al. (Nature Communication, 2019) also studied Cobb-Douglas Production functions.

Minor:

- What is the range of population in the 101 SUA?
- References are messed up, neither alphabetically, nor in the order of appearance.
- Adj. R2 is not written consistently.
- Is $X = \sum(X_i)$ at all used in the Cobb-Douglas Analysis? In the present manuscript it might be misleading.
- Cobb-Douglas Analysis is not shown for OCCP2. Despite the many categories it can still be included by using the aggregation from OCCP1, e.g. showing median, standard deviation, ranges or alike.
- Which exponent results form the conventional GDP vs. population scaling for the USA? The authors discuss it for Australia, why not USA?
- It would be nice to also have Figures.

Review form: Reviewer 2

Is the manuscript scientifically sound in its present form?

Yes

Are the interpretations and conclusions justified by the results?

Yes

Is the language acceptable?

Yes

Do you have any ethical concerns with this paper?

No

Have you any concerns about statistical analyses in this paper?

No

Recommendation?

Accept with minor revision (please list in comments)

Comments to the Author(s)

This paper is a nice piece of work that attempt at reconnecting scaling laws with economic elasticities. I think it is worth publishing it, provided that the authors accept to add a few precisions that will enable a better understanding of their contribution.

The authors are already known for their remarkable work on urban issues and scaling. They propose to compare scaling laws in Australia and USA regarding total number of workers and income when disaggregated by occupational categories.

Interestingly enough, the authors begin with a parallel between scaling laws and urban productivity measurement, referring to Cobb Douglas functions and their experimentation on diverse locations. Their intuition is that population size in itself is not such a good predictor of urban productivity whereas mix of industry and population skills could perform better.

In section 2 the authors analyse scaling relationship between income and population when income is weighted by the occupational structure of the population. This is applied on Australian urban units. The Australian nomenclature of occupational groups is presented in terms of granularity, but nothing is said about the major rationales of the classification: which are the criteria that differentiate the major categories of this nomenclature? To what extent is it comparable internationally? The question is important as we will show below.

The originality of the paper is in section 3 where the linkage between scaling laws and classical productivity functions is operated when disaggregating occupations according to their income level. This is interesting in theory. However, income is a partly redundant variable with the classification of occupational groups (as appear in table 2). Even if there are variations of income inside the occupational categories, the weighting adds little practical knowledge to the problem of the relationship with city size. Table 3 where scaling exponents are computed for number of workers in occupational groups relatively to income level in cities do not contribute much to our information, even if we share the author's conclusion about the necessity of sorting activities for understanding variations in productivity among cities.

Indeed, the conclusion regarding the values of exponents (high skill occupations scale supralinearly while lower skills scale sublinearly) rejoins a result that was already established in reference 11 by F. Paulus and C. Vacchiani-Marcuzzo for France and US cities (pp 237-260 (especially see tables 8.3 and 8.4 and figure 8.7)

The authors refer rightly to economic theory of urban productivity, they perhaps could mention how such results tend to confirm the geographical evolutionary theory of urban systems which connect the value of scaling exponents with the stage of activities in a hierarchical diffusion process of innovation waves within the system of cities. This would be in agreement with their conclusion about expecting future advances in research from investigations in "causal economic and physical processes that result in observed scaling behaviours".

The results presented on page 11 in this paper on the case of US cities could also be compared to those established (through comparing France and US) in another reference that is not quoted

here: Paulus F., & Vacchiani-Marcuzzo C. (2016). Knowledge industry and competitiveness: Economic trajectories of French urban areas (1962-2008). In A. Cusinato, A. Philippopoulos-Mihalopoulos, Knowledge-creating Milieus in Europe: Firm Cities, Territories. Springer, pp. 157-170.

Decision letter (RSOS-191638.R0)

11-Nov-2019

Dear Dr Sarkar,

On behalf of the Editors, I am pleased to inform you that your Manuscript RSOS-191638 entitled "Evidence for localization and urbanization economies in urban scaling" has been accepted for publication in Royal Society Open Science subject to minor revision in accordance with the referee suggestions. Please find the referees' comments at the end of this email.

The reviewers and handling editors have recommended publication, but also suggest some minor revisions to your manuscript. Therefore, I invite you to respond to the comments and revise your manuscript.

- Ethics statement

- Data accessibility

If you wish to submit your supporting data or code to Dryad (<http://datadryad.org/>), or modify your current submission to dryad, please use the following link:
<http://datadryad.org/submit?journalID=RSOS&manu=RSOS-191638>

- Competing interests

- Authors' contributions

- Acknowledgements

- Funding statement

Because the schedule for publication is very tight, it is a condition of publication that you submit the revised version of your manuscript before 20-Nov-2019. Please note that the revision deadline will expire at 00.00am on this date. If you do not think you will be able to meet this date please let me know immediately.

- 1) A text file of the manuscript (tex, txt, rtf, docx or doc), references, tables (including captions) and figure captions. Do not upload a PDF as your "Main Document";
- 2) A separate electronic file of each figure (EPS or print-quality PDF preferred (either format should be produced directly from original creation package), or original software format);
- 3) Included a 100 word media summary of your paper when requested at submission. Please ensure you have entered correct contact details (email, institution and telephone) in your user account;

- 4) Included the raw data to support the claims made in your paper. You can either include your data as electronic supplementary material or upload to a repository and include the relevant doi within your manuscript. Make sure it is clear in your data accessibility statement how the data can be accessed;
- 5) All supplementary materials accompanying an accepted article will be treated as in their final form. Note that the Royal Society will neither edit nor typeset supplementary material and it will be hosted as provided. Please ensure that the supplementary material includes the paper details where possible (authors, article title, journal name).

If your manuscript is newly submitted and subsequently accepted for publication, you will be asked to pay the article processing charge, unless you request a waiver and this is approved by Royal Society Publishing. You can find out more about the charges at <https://royalsocietypublishing.org/rsos/charges>. Should you have any queries, please contact openscience@royalsociety.org.

Kind regards,
Lianne Parkhouse
Editorial Coordinator
Royal Society Open Science
openscience@royalsociety.org

on behalf of Professor Matjaz Perc (Associate Editor) and Miles Padgett (Subject Editor)
openscience@royalsociety.org

Reviewer comments to Author:

Reviewer: 1

Comments to the Author(s)

Evaluation of "Evidence for localization and urbanization economies in urban scaling"

The authors investigate urban scaling in terms of occupation and corresponding income. Therefore various analysis have been performed mostly for Australia but also for the USA. First, total personal income vs. population leads to slightly super-linear scaling with an exponent 1.05 for Australia. Second, the number of works separated into various occupations is regressed versus city size, resulting in super-linear scaling for high-income occupations and linear or even sub-linear for other occupations (again for Australia). Last, a multi-linear regression in the spirit of Cobb-Douglas Production function is performed providing an estimated exponent for each occupation group. For Australia, the authors find that the exponent of non-working population is

0 and if replaced by the city population it is also close to zero. Moreover, in both cases the sum of the exponents is close 1. Consistent results are obtained for MSA in the USA, although not shown. The authors draw conclusions as summarized in the Abstract (essentially it questions increasing returns to scale of urban economies).

The paper is not only a nice read, but it is also of high relevance for the urban scaling community and for the empirical understanding of agglomeration economies. Therefore, I recommend publication but I also think it would be good to address some issues.

- I have concerns about the multivariate linear regression. First it is not detailed how it has been done methodologically. Second, probably all X_i are correlated which can cause trouble. Third, it is striking that the sum of the exponents is always so close to 1. Can there be a methodological reason? E.g. that the sum converges to 1 when many independent (correlated) variables are used?
- The authors mention the problem of zero numbers in the counts of workers, but how do they deal with it? Or are there no zero-entries, not even in OCCP2.
- The authors employ some clustering. Neither the purpose is explained nor the clustering itself becomes clear.
- As the income is a product of the number of workers, it is expected that the analysis of number of workers and income lead to similar results.
- The authors cite an unusual small number of papers. Although there is no lower limit, I still suggest to expand the citations. E.g. recently H.V. Ribeiro et al. (Nature Communication, 2019) also studied Cobb-Douglas Production functions.

Minor:

- What is the range of population in the 101 SUA?
- References are messed up, neither alphabetically, nor in the order of appearance.
- Adj. R^2 is not written consistently.
- Is $X = \sum(X_i)$ at all used in the Cobb-Douglas Analysis? In the present manuscript it might be misleading.
- Cobb-Douglas Analysis is not shown for OCCP2. Despite the many categories it can still be included by using the aggregation from OCCP1, e.g. showing median, standard deviation, ranges or alike.
- Which exponent results form the conventional GDP vs. population scaling for the USA? The authors discuss it for Australia, why not USA?
- It would be nice to also have Figures.

Reviewer: 2

Comments to the Author(s)

This paper is a nice piece of work that attempt at reconnecting scaling laws with economic elasticities. I think it is worth publishing it, provided that the authors accept to add a few precisions that will enable a better understanding of their contribution.

The authors are already known for their remarkable work on urban issues and scaling. They propose to compare scaling laws in Australia and USA regarding total number of workers and income when disaggregated by occupational categories.

Interestingly enough, the authors begin with a parallel between scaling laws and urban productivity measurement, referring to Cobb Douglas functions and their experimentation on diverse locations. Their intuition is that population size in itself is not such a good predictor of urban productivity whereas mix of industry and population skills could perform better.

In section 2 the authors analyse scaling relationship between income and population when income is weighted by the occupational structure of the population. This is applied on Australian urban units. The Australian nomenclature of occupational groups is presented in terms of granularity, but nothing is said about the major rationales of the classification: which are the criteria that differentiate the major categories of this nomenclature? To what extent is it comparable internationally? The question is important as we will show below.

The originality of the paper is in section 3 where the linkage between scaling laws and classical productivity functions is operated when disaggregating occupations according to their income level. This is interesting in theory. However, income is a partly redundant variable with the classification of occupational groups (as appear in table 2). Even if there are variations of income inside the occupational categories, the weighting adds little practical knowledge to the problem of the relationship with city size. Table 3 where scaling exponents are computed for number of workers in occupational groups relatively to income level in cities do not contribute much to our information, even if we share the author's conclusion about the necessity of sorting activities for understanding variations in productivity among cities.

Indeed, the conclusion regarding the values of exponents (high skill occupations scale supralinearly while lower skills scale sublinearly) rejoins a result that was already established in reference 11 by F. Paulus and C. Vacchiani-Marcuzzo for France and US cities (pp 237-260 (especially see tables 8.3 and 8.4 and figure 8.7)

The authors refer rightly to economic theory of urban productivity, they perhaps could mention how such results tend to confirm the geographical evolutionary theory of urban systems which connect the value of scaling exponents with the stage of activities in a hierarchical diffusion process of innovation waves within the system of cities. This would be in agreement with their conclusion about expecting future advances in research from investigations in "causal economic and physical processes that result in observed scaling behaviours".

The results presented on page 11 in this paper on the case of US cities could also be compared to those established (through comparing France and US) in another reference that is not quoted here: Paulus F., & Vacchiani-Marcuzzo C. (2016). Knowledge industry and competitiveness: Economic trajectories of French urban areas (1962-2008). In A. Cusinato, A. Philippopoulos-Mihalopoulos, Knowledge-creating Milieus in Europe: Firm Cities, Territories. Springer, pp. 157-170.

Author's Response to Decision Letter for (RSOS-191638.R0)

See Appendix A.

Decision letter (RSOS-191638.R1)

27-Feb-2020

Dear Dr Sarkar,

It is a pleasure to accept your manuscript entitled "Evidence for localization and urbanization economies in urban scaling" in its current form for publication in Royal Society Open Science. The comments of the reviewer(s) who reviewed your manuscript are included at the foot of this letter.

You can expect to receive a proof of your article in the near future. Please contact the editorial

office (openscience_proofs@royalsociety.org) and the production office (openscience@royalsociety.org) to let us know if you are likely to be away from e-mail contact -- if you are going to be away, please nominate a co-author (if available) to manage the proofing process, and ensure they are copied into your email to the journal.

on behalf of Professor Matjaz Perc (Associate Editor) and Miles Padgett (Subject Editor)
openscience@royalsociety.org

Appendix A

Dear Editor,

Thank you for the minor revisions decision, and for the helpful comments from the reviewers and the editorial team. Below, we provide our responses and record of amendments to our manuscript. Our response text is in blue, and the original comments are in grey. Our manuscript has the changes highlighted in yellow.

Best regards,
Authors.

11-Nov-2019

Dear Dr Sarkar,

On behalf of the Editors, I am pleased to inform you that your Manuscript RSOS-191638 entitled "Evidence for localization and urbanization economies in urban scaling" has been accepted for publication in Royal Society Open Science subject to minor revision in accordance with the referee suggestions. Please find the referees' comments at the end of this email.

The reviewers and handling editors have recommended publication, but also suggest some minor revisions to your manuscript. Therefore, I invite you to respond to the comments and revise your manuscript.

- Ethics statement

Our study does not include any humans or animals, and did not require any ethical approval. Thus, we are not including any ethics statement.

- Data accessibility

<http://datadryad.org/submit?journalID=RSOS&manu=RSOS-191638>

The data and code files related to this project are available at:

<https://github.com/SomwritaSarkar/UrbanScalingCode>

The data accessibility statement is now included in the paper at the end.

- Competing interests

The authors declare no competing interests. The statement is now included in the paper.

- Authors' contributions

The author contribution statement is now included in the manuscript.

- Acknowledgements

The acknowledgement section was already included. And has been amended to acknowledge comments from colleagues and anonymous referees.

- Funding statement

No funding sources. Statement included.

Because the schedule for publication is very tight, it is a condition of publication that you submit the revised version of your manuscript before 20-Nov-2019. Please note that the revision deadline will expire at 00.00am on this date. If you do not think you will be able to meet this date please let me know immediately.

1) Identifying all the changes that have been made (for instance, in coloured highlight, in bold text, or tracked changes);

Uploaded

Uploaded

1) A text file of the manuscript (tex, txt, rtf, docx or doc), references, tables (including captions) and figure captions. Do not upload a PDF as your "Main Document";

Uploaded

2) A separate electronic file of each figure (EPS or print-quality PDF preferred (either format should be produced directly from original creation package), or original software format);

N/A

3) Included a 100 word media summary of your paper when requested at submission. Please ensure you have entered correct contact details (email, institution and telephone) in your user account;

Uploaded

4) Included the raw data to support the claims made in your paper. You can either include your data as electronic supplementary material or upload to a repository and include the relevant doi within your manuscript. Make sure it is clear in your data accessibility statement how the data can be accessed;

Data and code repository link included.

5) All supplementary materials accompanying an accepted article will be treated as in their final form. Note that the Royal Society will neither edit nor typeset supplementary material and it will be hosted as provided. Please ensure that the supplementary material includes the paper details where possible (authors, article title, journal name).

N/A

Please let us know how these charges may be transferred from a University of Sydney research account to Royal Society.

If your manuscript is newly submitted and subsequently accepted for publication, you will be asked to pay the article processing charge, unless you request a waiver and this is approved by Royal Society Publishing. You can find out more about the charges at <https://royalsocietypublishing.org/rsos/charges>. Should you have any queries, please contact openscience@royalsociety.org.

Kind regards,

on behalf of Professor Matjaz Perc (Associate Editor) and Miles Padgett (Subject Editor)
openscience@royalsociety.org

Reviewer comments to Author:

Reviewer: 1
Comments to the Author(s)

Evaluation of "Evidence for localization and urbanization economies in urban scaling"

The authors investigate urban scaling in terms of occupation and corresponding income. Therefore various analysis have been performed mostly for Australia but also for the USA. First, total personal income vs. population leads to slightly super-linear scaling with an exponent 1.05 for Australia. Second, the number of works separated into various occupations is regressed versus city size, resulting in super-linear scaling for high-income occupations and linear or even sub-linear for other occupations (again for Australia). Last, a multi-linear regression in the spirit of Cobb-Douglas Production function is performed providing an estimated exponent for each occupation group. For Australia, the authors find that the exponent of non-working population is 0 and if replaced by the city population it is also close to zero. Moreover, in both cases the sum of the exponents is close 1. Consistent results are obtained for MSA in the USA, although not shown. The authors draw conclusions as summarized in the Abstract (essentially it questions increasing returns to scale of urban economies).

The paper is not only a nice read, but it is also of high relevance for the urban scaling community and for the empirical understanding of agglomeration economies. Therefore, I recommend publication but I also think it would be good to address some issues.

Our thanks to the reviewer for the positive comments. We respond to the points below, and include suggested amendments to the paper, highlighted in yellow.

- I have concerns about the multivariate linear regression. First it is not detailed how it has been done methodologically. Second, probably all X_i are correlated which can cause trouble. Third, it is striking that the sum of the exponents is always so close to 1. Can there be a methodological reason? E.g. that the sum converges to 1 when many independent (correlated) variables are used?

The multivariate linear regression has been carried out using the standard inbuilt Matlab multivariate regression package *fitlm* (fit linear model). We have included this information in the paper now (highlighted in yellow), and also the data and matlab code is included as part of the paper, so other researchers can run the analysis if they so desire.

We agree to the reviewer's second comment also: if the X_i are correlated, then the problem of multicollinearity can arise. However, typically the presence of multicollinearity does not reduce the predictive power or reliability of the results from the model as a whole, but nothing can be said about the predictive power of the individual variables. In this case, our main aim has been to show that overall linear scaling emerges when the occupational and industrial variables are used. We therefore do not make any claims in the paper on the explanatory power of the individual variables, we only note that as a whole the model emerges as a linear scaling model. The suggestion of correlated variables is, however, very interesting – we thank the reviewer for bringing this to our notice, and we will definitely explore this in future work.

Third, we have not been able to find a methodological reason behind the idea that if there are several correlated variables, the sum will always converge to 1 when correlated variables are used. Infact, several examples could be found by authors where the beta could sum to more than 1, or less than 1. We will explore this too in our future research.

Overall, we note that our results are reliable in that we have followed the standard procedures for this type of analysis prevalent in the urban scaling literature in general, and have also tested reliability by comparing the linear regression results with the MLE methods, suggested as a test by authors of Leitao2016.

- The authors mention the problem of zero numbers in the counts of workers, but how do they deal with it? Or are there no zero-entries, not even in OCCP2.

The reviewer is correct – at the OCCP2 level, in general, the granularity is enough to ensure that there are no zero counts. In the odd case that there was a zero count, the standard procedure followed was to assume 1 worker, so that the log value comes to zero.

- The authors employ some clustering. Neither the purpose is explained nor the clustering itself becomes clear.

The authors employ clustering in order to cluster the various occupations into broadly high, low, and middle income occupations. The reasons for this have been better explained in the manuscript now (highlighted in yellow).

- As the income is a product of the number of workers, it is expected that the analysis of number of workers and income lead to similar results.

While this is largely true, a high income occupation could have either a low or a high number of workers in a particular city, or a low income occupation could have either a low or a high number of workers in a particular city, and this could then lead to different signatures for income and workers. Therefore, we have explored both and reported results on both.

- The authors cite an unusual small number of papers. Although there is no lower limit, I still suggest to expand the citations. E.g. recently H.V. Ribeiro et al. (Nature Communication, 2019) also studied Cobb-Douglas Production functions.

Thanks for the suggestion – we have now included this paper in our reference list.

Minor:

- What is the range of population in the 101 SUA?

This information is now included in the paper (highlighted in yellow).

- References are messed up, neither alphabetically, nor in the order of appearance.

Thanks for the observation – we have now turned the reference list alphabetical.

- Adj. R2 is not written consistently.

Thanks for noting – we have repaired this.

- Is $X = \sum(X_i)$ at all used in the Cobb-Douglas Analysis? In the present manuscript it might be misleading.

We are not entirely clear on this comment. We have checked – and to the best of our understanding – the mathematically stated Cobb-Douglas form and the $\sum(X_i)$ are used to draw the conclusions of the linear scaling.

- Cobb-Douglas Analysis is not shown for OCCP2. Despite the many categories it can still be included by using the aggregation from OCCP1, e.g. showing median, standard deviation, ranges or alike.

The main results from the Cobb-Douglas analysis at the OCCP2 level are included in the paper, but we feel that simply reporting the detailed set of numbers did not add any new insight or information to the conclusions already being discussed. We have though conducted the full experiments for scientific integrity, and the data and code is included, so that if other researchers wish they can re-run the experiments.

- Which exponent results form the conventional GDP vs. population scaling for the USA? The authors discuss it for Australia, why not USA?

We do indeed discuss this in the paper – at the end of page 5 – we compare the conventional results for the USA with our results.

- It would be nice to also have Figures.

We felt that all the main information we wanted to convey could indeed be represented in figures visually, but we decided to present tables so that all the numerical information is available to directly read – we felt that this was a better mode of presenting the results.

Reviewer: 2

Comments to the Author(s)

This paper is a nice piece of work that attempt at reconnecting scaling laws with economic elasticities. I think it is worth publishing it, provided that the authors accept to add a few precisions that will enable a better understanding of their contribution.

Thanks for the positive comments and suggestions – we discuss where we have included the amendments.

The authors are already known for their remarkable work on urban issues and scaling. They propose to compare scaling laws in Australia and USA regarding total number of workers and income when disaggregated by occupational categories.

Interestingly enough, the authors begin with a parallel between scaling laws and urban productivity measurement, referring to Cobb Douglas functions and their experimentation on diverse locations. Their intuition is that population size in itself is not such a good predictor of urban productivity whereas mix of industry and population skills could perform better.

In section 2 the authors analyse scaling relationship between income and population when income is weighted by the occupational structure of the population. This is applied on Australian urban units. The Australian nomenclature of occupational groups is presented in terms of granularity, but nothing is said about the major rationales of the classification: which are the criteria that differentiate the major categories of this nomenclature? To what extent is it comparable internationally? The question is important as we will show below.

The originality of the paper is in section 3 where the linkage between scaling laws and classical productivity functions is operated when disaggregating occupations according to their income level. This is interesting in theory. However, income is a partly redundant variable with the classification of occupational groups (as appear in table 2). Even if there are variations of income inside the occupational categories, the weighting adds little practical knowledge to the problem of the relationship with city size. Table 3 where scaling exponents are computed for number of workers in occupational groups relatively to income level in cities do not contribute much to our information, even if we share the author's conclusion about the necessity of sorting activities for understanding variations in productivity among cities.

For the reviewer the question of the variations between the occupational groups and how it related to city size is important. However, the main message of our work is that, yes, there are indeed variations, as the reviewer very rightly suggests, but on the whole, given all of those variations, across different occupational classifications across the USA and Australia (whether or not these compare – the fact that they are different may indeed add to the robustness of our results) – with all the above given, we observe a linear (and not superlinear) scaling. That was the main message in the paper, that we hope to convey. It appears that the reviewer indeed agrees with this observation.

Indeed, for us too, the necessity of exploring the sorting of activities for understanding variations in productivity among cities (in which we begin to explore the individual explanatory power of each of the occupational groups) is an area of future work resulting from the work done in this paper.

Indeed, the conclusion regarding the values of exponents (high skill occupations scale supralinearly while lower skills scale sublinearly) rejoins a result that was already established in reference 11 by F. Paulus and C. Vacchiani-Marcuzzo for France and US cities (pp 237-260 (especially see tables 8.3 and 8.4 and figure 8.7)

We are glad that the reviewer agrees that our findings agree with previous findings from other studies focused on other regions in the world (e.g. France).

The authors refer rightly to economic theory of urban productivity, they perhaps could mention how such results tend to confirm the geographical evolutionary theory of urban systems which connect the value of scaling exponents with the stage of activities in a hierarchical diffusion process of innovation waves within the system of cities. This would be in agreement with their conclusion about expecting future advances in research from investigations in “causal economic and physical processes that result in observed scaling behaviours”.

Indeed, we already note this in the paper – we include several Pumain et al references, and on page 3 we say: In particular, some authors have showed that the maturity of the specific industry is a more relevant determinant for the exponent than population size [13-16]. We have added now the point on hierarchical diffusion (highlighted in yellow).

The results presented on page 11 in this paper on the case of US cities could also be compared to those established (through comparing France and US) in another reference that is not quoted here: Paulus F., & Vacchiani-Marcuzzo C. (2016). Knowledge industry and competitiveness: Economic trajectories of French urban areas (1962-2008). In A. Cusinato, A. Philippopoulos-

Mihalopoulos, Knowledge-creating Milieus in Europe: Firm Cities, Territories. Springer, pp. 157-170.

Thanks for this suggestion – we have now included this reference in the paper, as part of the geographical evolutionary theory literature.

Journal Name: Royal Society Open Science

Journal Code: RSOS

Online ISSN: 2054-5703

Journal Admin Email: openscience@royalsociety.org

Journal Editor: Andrew Dunn

Journal Editor Email: openscience@royalsociety.org

MS Reference Number: RSOS-191638

Article Status: SUBMITTED

MS Dryad ID: RSOS-191638

MS Title: Evidence for localization and urbanization economies in urban scaling

MS Authors: Sarkar, Somwrita; Arcaute, Elsa; Hatna, Erez; Alizadeh, Tooran; Searle, Glen; Batty, Michael

Contact Author: Somwrita Sarkar

Contact Author Email: somwrita.sarkar@sydney.edu.au

Contact Author Address 1:

Contact Author Address 2:

Contact Author Address 3:

Contact Author City:

Contact Author State:

Contact Author Country: United Kingdom of Great Britain and Northern Ireland

Contact Author ZIP/Postal Code:

Keywords: Agglomeration economies, Urban scaling, Urbanization and localization economies

Abstract: We study the scaling of (i) numbers of workers and aggregate incomes by occupational categories against city size, and (ii) total incomes against numbers of workers in different occupations, across the functional metropolitan areas of Australia and the US. The number of workers and aggregate incomes in specific high income knowledge economy related occupations and industries show increasing returns to scale by city size, showing that localization economies within particular industries account for superlinear effects. However, when total urban area incomes and/or Gross Domestic Products are regressed using a generalised Cobb-Douglas function against the number of workers in different occupations as labour inputs, constant returns to scale in productivity against city size are observed. This implies that the urbanization economies at the whole city level show linear scaling or constant returns to scale. Furthermore, industrial and occupational organisations, not population size, largely explain the observed productivity variable. The results show that some very specific industries and occupations contribute to the observed overall superlinearity. The findings suggest that it is not just size but also that it is the diversity of specific intra-city organization of economic and social activity and physical infrastructure that should be used to understand urban scaling behaviors.

EndDryadContent